# Magnitude, trends and determinants of skilled delivery from Kilite-Awlaelo Health Demographic Surveillance System, Northern Ethiopia, 2009- 2017

Haftom Temesgen Abebe[1]*, Mache Tsadik Adhana[2], Mengistu Welday Gebremichael[3], Kebede Embaye Gezae[1‡], Assefa Ayalew Gebreslassie[2‡]

1 Department of Biostatistics, College of Medicine and Health Science, Mekelle University, Mekelle, Ethiopia, 2 Department of Reproductive Health, College of Medicine and Health Science, Mekelle University, Mekelle, Ethiopia, 3 School of Midwives, College of Medicine and Health Science, Mekelle University, Mekelle, Ethiopia

☯ These authors contributed equally to this work.
‡ These authors also contributed equally to this work.
* haftoma@gmail.com

**Data Availability Statement:** The data has potentially identifying information including name and households number. The data can be obtained

## Abstract

### Background

The fundamental approach to improve maternal and neonatal health is increasing skilled delivery rate. Women giving birth at health institutions can prevent maternal and neonatal deaths by getting skilled birth attendance. In Ethiopia, despite a significant decrease in maternal mortality over the past decade, still a significant number of women give birth at home. Moreover, evidence from population-based longitudinal studies on skilled delivery is limited. Therefore, this study aims to investigate the magnitude, trend, and determinants of skilled delivery in Kilite-Awlaelo Health Demographic Surveillance System (KA-HDSS), Northern Ethiopia.

### Method

Population-based longitudinal study design was conducted by extracting data for nine consecutive years (2009–2017) from KA-HDSS database. In order to measure the trends of skilled delivery, KA-HDSS data sets were analyzed (2009–2017). Bivariate and multivariate analyses were performed using STATA version 16. A multivariable binary logistic regression model was fitted to assess determinants of skilled delivery and odds ratio with 95% CI was used to assess presence of associations at a 0.05 level of significance.

### Results

The skilled delivery rate have continuously increased among reproductive age women from 15.12% (95% CI: 13.30% - 17.09%) in 2010 to 95.85% (95% CI: 94.58% - 96.895%) in 2017. The skilled delivery rate becomes high (> = 82) in the period of 2014–2017. Education, residence, marital status, occupation and antenatal care (ANC) visits were the most

from the institutional office Kilite-Awlaelo Health Demographic Surveillance System (KA-HDSS), College of Heath Science, Mekelle University, Email: ka.hdss.2011@gmail.com; Tel: +251914743841.

**Funding:** The author(s) received no specific funding for this work.

**Competing interests:** The authors have declared that no competing interests exist.

**Abbreviations:** KA-HDSS, Kilite-Awlaelo Health and Demographic Surveillance System; AOR, Adjusted odds ratio; CI, Confidence interval; ANC, antenatal care; SDG, Sustainable Development Goals; EDHS, Ethiopian Demographic and Health Surveys; MMR, Maternal Mortality Ratio; EPMM, ending preventable maternal mortality; IRB, Institutional Review Board.

important determinants for skilled delivery among reproductive age women during the period of high skilled delivery rate (2014–2017). Women urban dwellers had about 28 times (AOR = 27.66; 95% CI: 3.86–196.97) higher odds to deliver by skilled birth attendants than rural dwellers. Unmarried women who gave birth were 2.18 (AOR: 2.18; 95% CI: 1.30–3.64) times more likely to have skilled delivery service compared to those married. Likewise, women with four or more ANC visits were 3.2 times more likely to undergo skilled delivery service than those having no ANC visits (AOR: 3.16; 95% CI: 2.33–4.28). Moreover, women having at least a secondary education were 2 times more likely to have skilled delivery service compared to those women with no formal education (AOR = 2.10, 95% CI: 1.18–3.74).

## Conclusion

Regardless of the importance of health facility delivery, a significant number of women still deliver at home attended by unskilled birth attendants. There has been a substantial increase in use of health facilities for delivery among women in the reproductive age. The factors affecting skilled delivery among reproductive age women were educational level, residence, marital status, occupation and use of ANC service. Maternal health related interventions are needed to change women's attitudes towards skilled delivery. Moreover, ANC coverage should be increased to improve skilled delivery service.

## Introduction

The skilled attendant is a health professional who may be a midwife, doctor, or nurse with midwifery and life-saving skills [1]. Skilled health personnel are competent maternal and newborn health professionals who are educated, trained and qualified based on national and international standards [2]. An estimated 289,000 women die per annum due to complications related to pregnancy and childbirth [3]. Two thirds of maternal deaths occur after delivery [4, 5]. Poor maternal and neonatal care results in 2.9 million neonatal deaths every year [6]. Of those global deaths 99% occurred in low- and middle-income countries including Ethiopia [7, 8]. The lifetime risk of dying in pregnancy situations are 1 in 30,000 in Sweden and 1 in 16 in sub-Saharan Africa [9].

The fundamental approach to improve maternal and neonatal health is increasing skilled delivery rate [10]. The demonstrated strategies to diminish maternal mortality are skilled birth attendance, referral for complications and universal availability of emergency obstetric care, such as Caesarian section [11]. Sustainable Development Goals (SDGs) goal three aims to decrease the global maternal mortality ratio to less than 70 per 100,000 live births and ensure universal access to sexual and reproductive health services by 2030. Despite the availability of access to healthcare service, the quality of care matters in the outcome of pregnancy.

Ethiopia has good progress in reducing maternal mortality. The Ethiopian Demographic and Health Surveys (EDHS) showed that Maternal Mortality Ratio (MMR) has dropped from 871 in 2000 to 676 in 2011 and then to 412 in 2016 per 100,000 live births, which is in line with the findings of the UN Inter-Agency Group (UN-IAG) that showed that the MMR had declined from 1,400 to 353 maternal deaths per 100,000 live births. The absolute number of women who died during pregnancy or childbirth had also decreased by nearly 75%, from 31,000 to around 11,000 from 1990 to 2015. [9, 10, 12, 13].

According to the EDHS report, the skilled delivery of the three surveys showed that 6% in 2005, 11% in 2011 and 28% in 2016. The skilled delivery of those surveys in Tigray were 6% in

2005, 12% in 2011 and 69% in 2016. This lags behind the health sector transformation plan of the country, which was set to be 90% [14–17].

Achieving "proportion of births attended by skilled health personnel" is Sustainable Development Goal 3. This requires strong and effective strategies, and accurate measurement and monitoring of progress for ending preventable maternal mortality [18]. Despite significant reduction in maternal mortality in the past decade, ending preventable maternal mortality (EPMM) remains an unfinished agenda and one of the world's most critical challenges [18].

In the light of this evidence, we have assessed Kilite-Awlaelo Health and Demographic Surveillance System (KA-HDSS) source data base of Mekelle University College of Health Sciences, collected from 2009 to 2017, to understand the size and range of changing delivery trends in skilled delivery attendance [19]. So far few studies in Ethiopia have been conducted regarding skilled delivery [20–23]. Evidence from population based longitudinal studies on skilled delivery is limited. Besides, there is no information regarding the trend of skilled delivery over time. Thus, the study aimed to assess the magnitude, trend and determinants of skilled delivery using population based longitudinal data from KA-HDSS in Northern Ethiopia.

## Materials and methods

### Study setting and design

KA-HDSS is an ongoing open cohort study, located in Northern Ethiopia and hosted by Mekelle University. The site has three climatic zones which includes lowland, midland and high land. Administratively, it was established in 9 rural and 1 urban kebelles in April 2009 (a kebelle is the smallest administrative component in the country). At the beginning of the surveillance, baseline socio-demographic characteristics of 65,848 individuals living in 14,455 households were collected through a census. At the same time, a unique surveillance identification number was given to every enumerated cohort and household to facilitate linking information during longitudinal observation. In 2016, 2 urban kebelles were added as part of the study area and the number of household increased to 21,688. In 2017, the project has made 11 updates rounds with population of 101,146 living in 21,688 households in 12 kebeles (9 rural and 3 urban). A house to house visit is done to capture information regarding individuals, pregnancy observation, pregnancy outcomes, deaths, births and migration. Events are collected as it occurs and updated every six months [19].

### Data sources and study population

The source of data for this study was from KA-HDSS. The study population for this study was all women who had at least one birth in KA-HDSS from April, 2009 to December, 2017.

### Data extraction tool and study variables

Data regarding the skilled delivery were extracted mainly from pregnancy observation, pregnancy outcome, and relationship tables of KA-HDSS data considering the relevance of each explanatory variable on the prediction of skilled delivery rate in the population.

**Dependent variable.** The dependent variable in this study was skilled delivery. It was a dichotomized response as 1 if a woman gave birth by skilled birth attendants and 0 otherwise (if a woman gave birth by unskilled birth attendants).

**Independent variables.** The independent variables were classified as socio-demographic variables, and pregnancy outcome and related variables.

The socio-demographic variables are age, ethnicity, religion, marital status, occupation, level of education, and place of residence. The pregnancy outcome and related variables are

age at pregnancy, number of ANC visits, bed net use, number of children born alive, number of children dead, number of previous pregnancy and previous pregnancy outcome.

## Statistical data analysis

Data were cleaned and analysed in STATA version 15 statistical tool. The study population were described using frequency (percentage), mean (±standard deviation (sd)) depending on the nature of data (variables). A line graph was used to observe the trend of institutional delivery (number of skilled deliveries per 100) over time. Moreover, a cross-tabulation between each categorical independent variable and the outcome variable was done to check whether the expected cell counts were adequate or not. Besides, descriptive statistics, a rigorous statistical method was applied to identify the determinants of delivery in the study setting. Bivariate analysis was performed to assess the relationship between the dependent and independent variables.

A multivariable binary logistic regression analysis was fitted to identify the adjusted effect of each determinant on the skilled delivery among the study population of the specified study setting. The assumptions of multicollinearity between two or more independent variables were checked. Goodness of fit of the model was assessed using Hosmer-Lemeshow test. Decision regarding the statistical significance effect of independent variables on skilled delivery was made based on either the 95% CIs for AOR or associated P-values.

## Ethical approval and consent to participate

Permission to access the data was obtained from Mekelle University KA HDSS via an agreement on the data sharing policy after ethical approval was obtained from Institutional Review Board (IRB) of Mekelle University, College of Health Sciences. Consent to participate was fully waived as the study participants were not directly involved in the study (i.e. an already existing data were utilized for analysis in the current study). Moreover, the confidentiality of data was kept as there were no personal identifiers used and neither the raw data nor the extracted data were passed to a third person (i.e. it is only used for the purpose of the study).

## Results

### Socio-demographic characteristics

Overall, 7,263 women were included in the study for a total of 11,925 observations for delivery in the last nine years (2009–2017). Of the 7,263 women, 3,842(52.89%) were Para I, 2,320 (31.94%) were Para II, 966(13.30%) were Para III, 130(1.79%) were Para IV and 5(0.07%) were Para V. The mean and standard deviation of the women's age at pregnancy who were included in the study was found to be 30.12±6.8 years. Almost all (99.3%) of the participants were Tigreans, and 7,164(98.6%) were also orthodox religion followers. Regarding their occupation, more than half (56.73%) were house wives. Moreover, 4,390(60.4%) of women had no formal education, and more than two thirds (68.9%) were married, 6,466(89.0%) were rural dwellers (Table 1).

### Maternal health service utilization characteristics

Based on their ANC visits, 8,515 of 11,925 (71.40%) of pregnancies had at least one ANC visit. Of the women who had ANC visits, 2,234(26.24%) reached ANC 4 and above. Based on pregnancy outcomes, of the total observations, 11,593(97.22%), 178(1.49%) and 154(1.29%) had live births, abortion and still births, respectively. The median frequency of ANC visit was 2.5.

**Table 1. Socio-demographic characteristics of the women who delivery in the last nine years, Kilite-Awlaelo HDSS site, Northern Ethiopia, 2009–2017.**

| Socio demographic | Frequency (%) | Skilled delivery | | $X^2$ test |
|---|---|---|---|---|
| | | Yes (%) | No (%) | |
| **Residence of women (n = 7,263)** | | | | |
| Rural | 6,466(89.03) | 3,693(57.11) | 2,773(42.89) | <0.001 |
| Urban | 797(10.97) | 760(95.36) | 37(4.64) | |
| **Ethnicity women (n = 7,263)** | | | | |
| Tigray | 7,214(99.33) | 4,415(61.20) | 2,799(38.80) | 0.030 |
| Amhara | 16(0.22) | 15(93.75) | 1(6.25) | |
| Oromo | 31(0.43) | 21(67.74) | 10(32.26) | |
| Other | 2(0.03) | 2(100.00) | 0(0.0) | |
| **Religion (n = 7, 263)** | | | | |
| Orthodox | 7,164(98.64) | 4,370(61.00) | 2,794(39.00) | <0.001 |
| Muslim | 96(1.32) | 81(84.38) | 15(15.62) | |
| Catholic | 3(0.04) | 2(66.67) | 1(33.33) | |
| **Maternal Education(n = 7,263)** | | | | |
| Illiterate | 4,390(60.44) | 2,244(51.12) | 2,146(48.88) | <0.001 |
| Primary education | 2,109(29.04) | 1,573(74.58) | 536(25.42) | |
| Secondary education | 667(9.18) | 547(82.01) | 120(17.99) | |
| College and above | 97(1.34) | 89(91.75) | 8(8.25) | |
| **Maternal Occupation(n = 7,263)** | | | | |
| House wife | 4,120(56.73) | 2,051(49.78) | 2,069(50.22) | <0.001 |
| Farmer | 514(7.08) | 313(60.90) | 201(39.10) | |
| Student | 1,087(14.97) | 885(81.42) | 202(19.58) | |
| Merchant | 481(6.62) | 423(87.94) | 58(12.06) | |
| Government employee | 162(2.23) | 142(87.65) | 20(12.35) | |
| Daily laborer | 442(6.09) | 296(66.97) | 146(33.03) | |
| Unemployed | 129(1.78) | 100(77.52) | 29(22.48) | |
| Other | 328(4.52) | 243(74.09) | 85(25.91) | |
| **Marital status (n = 7,263)** | | | | |
| Married | 5,004(68.90) | 2,733(54.62) | 2,271(45.38) | <0.001 |
| Unmarried | 1,724(23.74) | 1,348(78.19) | 376(21.91) | |
| Others | 535(7.37) | 372(69.53) | 163(30.47) | |
| **Age at pregnancy (n = 11,925)** | | | | |
| 15–19 | 584(4.90) | 369(63.18) | 215(36.82) | <0.001 |
| 20–24 | 2,304(19.32) | 1,393(60.46) | 911(39.54) | |
| 25–29 | 2,738(22.96) | 1,770(64.65) | 968(35.35) | |
| 30–34 | 2,931(24.58) | 1,856(63.32) | 1,075(36.68) | |
| 35–39 | 2,233(18.73) | 1,428(63.95) | 805(36.05) | |
| 40–44 | 969(8.13) | 656(67.70) | 313(32.30) | |
| 45–49 | 166(1.39) | 121(72.89) | 45(27.11) | |

Regarding gravidity, 2,567(35.34%) of women had history of 5 and above pregnancies (Table 2).

Of the total deliveries, 7,535(63.19%) and 4,280 (35.89%) women delivered at health facilities and at home, respectively (Fig 1).

Of the total home deliveries, 97.87% were delivered by unskilled birth attendants. Of these 2,474(57.80%) deliveries were assisted by untrained traditional birth attendants, 390(9.11%) by

**Table 2. Maternal health service utilization characteristics of the women who delivery in the last nine years, Kilite-Awlaelo HDSS site, Northern Ethiopia, 2009–2017.**

| Health services | Frequency (%) | Skilled delivery | | $X^2$ test |
|---|---|---|---|---|
| | | Yes (%) | No (%) | |
| **ANC attendance (n = 11,925)** | | | | |
| Yes | 8,515(71.40) | 5,853(68.74) | 2,662(31.26) | <0.001 |
| No | 3,410(28.60) | 1,740(51.03) | 1,670(48.97) | |
| **Number of ANC (n = 8,515)** | | | | |
| ANC one visit | 2,400(28.19) | 1,521(63.38) | 879(36.63) | <0.001 |
| ANC two visit | 2,154(25.30) | 1,260(58.50) | 894(42.50) | |
| ANC three visit | 1,727(20.28) | 1,136(65.78) | 591(34.22) | |
| ANC 4 and above | 2,234(26.24) | 1,936(86.58) | 298(13.44) | |
| **Pregnancy outcomes (n = 11,925)** | | | | |
| Live births | 11,593(97.22) | 7,366(63.54) | 4,227(36.46) | 0.032 |
| Abortion | 178(1.49) | 130(73.03) | 48(26.97) | |
| Still births | 154(1.29) | 97(62.99) | 57(37.01) | |
| **Number of single and multiple tons (n = 11,593)** | | | | |
| Single tons | 11,447(98.74) | 7,261(63.43) | 4,186(36.57) | 0.090 |
| Multiple tons | 146(1.26) | 105(71.92) | 41(28.08) | |
| Gravidity (n = 7,263) | | | | |
| 1 | 1,770(24.37) | 1,366(77.18) | 404(22.82) | <0.001 |
| 2–4 | 2,926(40.29) | 1,856(63.43) | 1,070(36.57) | |
| > = 5 | 2,567(35.34) | 1,231(47.95) | 1,336(52.05) | |
| **Slept in bed net (n = 11,925)** | | | | |
| Yes | 5,299(44.44) | 3,232(60.99) | 2,067(39.01) | <0.001 |
| No | 3,575(29.98) | 2,364(66.13) | 1,211(33.87) | |
| Do not have | 3,051(25.58) | 1,997(65.45) | 1,054(34.55) | |

health extension workers, 352 (8.22%) by mother herself and significant number of mothers (16.33%) were assisted by others (neighbors, grandmother, mother-in-law) (Fig 2).

## Magnitude and trends of skilled delivery and ANC attendance

The trend of skilled delivery over the study period (2009–2017) significantly increased from 17.30% (95% CI: 13.12% -22.17%) in 2009 to 95.85% (95% CI: 94.58%-96.89%) in 2017. The

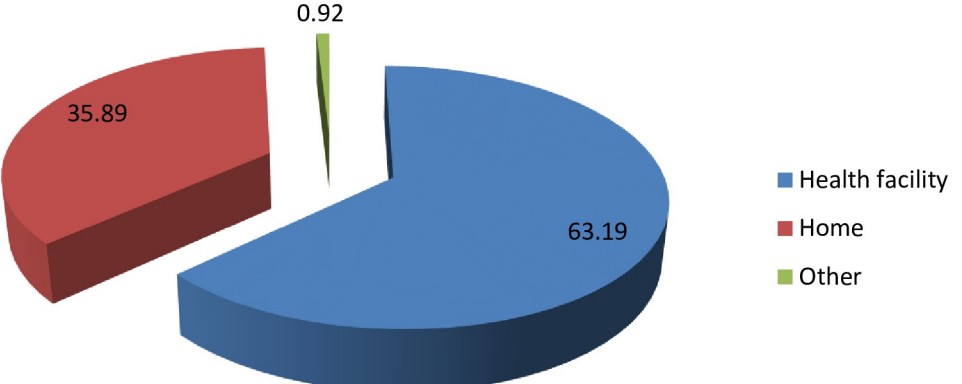

**Fig 1. Place of delivery from 2009–2017 KA-HDSS sites of Mekelle University College of Health Science, Mekelle, Tigray Ethiopia.**

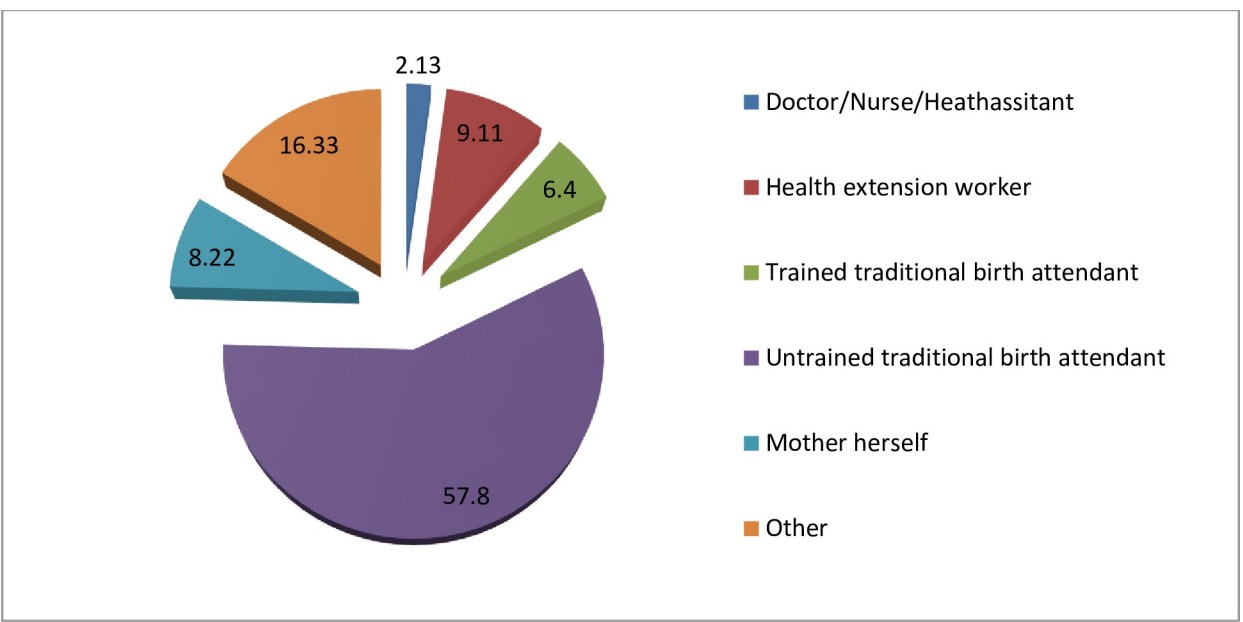

**Fig 2. Attendants of home delivery from 2009–2017 KA-HDSS sites, Mekelle University College of Health Science, Mekelle, Tigray Ethiopia.**

highest increment of skilled delivery was observed in the period 2011–2012 with a 33.35% increase followed by the period 2012–2013, which increased by 24.42% (Fig 3).

The trend of ANC attendance at least once in the study period (2009–2017) showed a significant change, increased from 48.44% (95% CI: 42.72%-54.21%) in 2009 to 94.7% (95% CI:

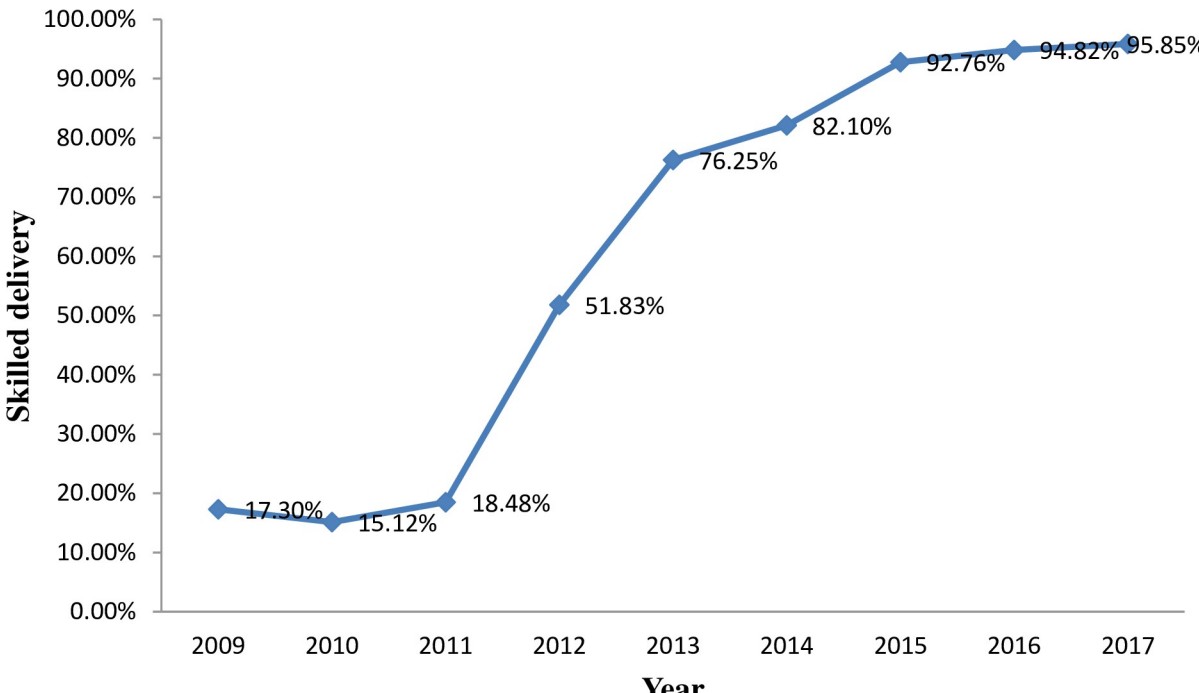

**Fig 3. Trends of skilled delivery from 2009–2017 KA-HDSS sites of Mekelle University College of Health Science, Mekelle, Tigray Ethiopia.**

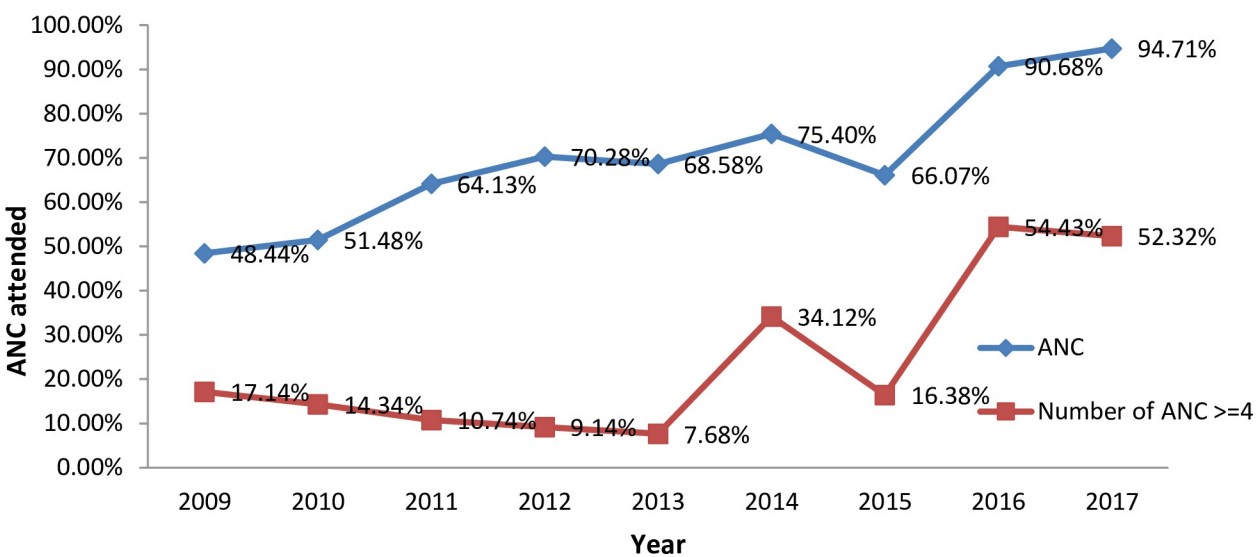

**Fig 4. Trends of ANC attendance from of 2009–2017 KA-HDSS sites of Mekelle University College of Health Science, Mekelle, Tigray Ethiopia.**

93.31–95.83%) in 2017. As shown in the Fig 4., ANC attendance 4+ continuously declined from 2009 to 2013 followed by inconsistent trend.

## Determinants of health facility deliveries

In the Bivariate analysis the variables residence, education, occupation, marital status, age at pregnancy, ANC attendance and number of ANC visits had statistically significant association with skilled delivery. The ANC attendance was not included in the multivariable binary logistic regression model due to multicollinearty with the number of ANC visits. Table 3 shows the determinants of skilled delivery among reproductive age women who gave birth during the period of 2009–2011 (when the skilled delivery rate was low) and 2014–2017 (when the skilled delivery rate was high). In the multivariable binary logistic regression model, the variables residence, education, occupation, marital status and number of ANC visits were found to be statistically significant contributors to skilled delivery during the period of 2014–2017. The Hosmer-Lemeshow test results confirmed that the model was a good fit for the data ($X^2(5)$ = 7.55, p-value = 0.1827). Keeping the effect of other predictors constant, unmarried women who gave birth during the period of high skilled delivery rate (2014 to 2017) were 2.18 (AOR: 2.18; 95% CI: 1.30–3.64) times more likely to have skilled delivery service compared to those who were married. In addition, women who had at least secondary, education were 2.10 times more likely to have skilled delivery service compared to those women with no education (AOR = 2.10, 95% CI: 1.18–3.74). Moreover, women with four or more ANC visits were 3.2 times more likely to have skilled delivery service than those having no ANC visits (AOR: 3.16; 95% CI: 2.33–4.28) (Table 3).

## Discussion

The essential approach to improve maternal and neonatal health is increasing skilled delivery rate. Women giving birth at health institutions can prevent maternal and neonatal deaths through getting skilled birth attendance, drugs to address labour complications and referrals to more advanced health institutions [12]. This study aimed to assess the magnitude, trend

**Table 3. Determinants of skilled delivery among reproductive age women who gave their recent birth from 2009 to 2011 and from 2014 to 2017 in KA-HDSS site, Tigray, Northern Ethiopia.**

| Variable | | 2009–2011 | | | | 2014–2017 | | | |
|---|---|---|---|---|---|---|---|---|---|
| | | AOR | SE | P-value | 95% CI | AOR | SE | P-value | 95% CI |
| **Residence** | Rural | 1.00 (reference category) | | | | 1.00 (reference category) | | | |
| | Urban | 35.46 | 7.63 | <0.001 | 23.26–54.06 | 27.58 | 27.66 | 0.001 | 3.86–196.97 |
| **Occupation** | Farmer | 1.00 (reference category) | | | | 1.00 (reference category) | | | |
| | Merchant | 2.25 | 0.98 | 0.061 | 0.97–5.26 | 0.60 | 0.20 | 0.121 | 0.31–1.15 |
| | Government employee | 5.95 | 2.76 | <0.001 | 2.40–14.78 | 0.79 | 0.52 | 0.719 | 0.22–2.87 |
| | Daily laborer | 1.03 | 0.34 | 0.936 | 0.54–1.95 | 0.44 | 0.16 | 0.021 | 0.22–0.88 |
| | Housewife | 0.71 | 0.19 | 0.181 | 0.42–1.18 | 0.55 | 0.15 | 0.027 | 0.33–0.94 |
| | Student | 1.17 | 0.40 | 0.652 | 0.60–2.27 | 0.34 | 0.13 | 0.005 | 0.16–0.72 |
| | Unemployed | 0.51 | 0.22 | 0.110 | 0.22–1.17 | 1.78 | 1.89 | 0.588 | 0.22–14.22 |
| | Other | 1.29 | 0.43 | 0.448 | 0.67–2.48 | 0.30 | 0.10 | <0.001 | 0.15–0.59 |
| **Marital Status** | Married | 1.00 (reference category) | | | | 1.00 (reference category) | | | |
| | Unmarried | 2.20 | 0.56 | 0.002 | 1.34–3.61 | 2.18 | 0.57 | 0.003 | 1.30–3.64 |
| | Others [a] | 1.14 | 0.35 | 0.669 | 0.63–2.08 | 1.50 | 0.46 | 0.182 | 0.83–2.72 |
| **Educational Status** | Illiterate | 1.00 (reference category) | | | | 1.00 (reference category) | | | |
| | Primary | 1.70 | 0.29 | 0.002 | 1.22–2.36 | 1.63 | 0.24 | 0.001 | 1.23–2.17 |
| | Secondary and above | 2.67 | 0.64 | <0.001 | 1.67–4.27 | 2.10 | 0.62 | 0.011 | 1.18–3.74 |
| **ANC visits** | No ANC attendance | 1.00 (reference category) | | | | 1.00 (reference category) | | | |
| | 1–3 ANC visits | 0.94 | 0.12 | 0.648 | 0.74–1.21 | 1.20 | 0.14 | 0.111 | 0.96–1.50 |
| | At least 4 ANC visits | 1.61 | 0.34 | 0.024 | 1.07–2.45 | 3.16 | 0.49 | <0.001 | 2.33–4.28 |
| **Previous pregnancy** | No | 1.00 (reference category) | | | | 1.00 (reference category) | | | |
| | Yes | 0.60 | 0.11 | 0.005 | 0.42–0.86 | 0.86 | 0.14 | 0.344 | 0.63–1.17 |
| **Age at pregnancy** | 15–19 | 1.00 (reference category) | | | | 1.00 (reference category) | | | |
| | 20–24 | 1.82 | 0.51 | 0.032 | 1.05–3.14 | 0.84 | 0.27 | 0.589 | 0.46–1.57 |
| | 25–29 | 2.40 | 0.72 | 0.004 | 1.33–4.33 | 1.11 | 0.38 | 0.758 | 0.57–2.19 |
| | 30–34 | 3.06 | 0.94 | <0.001 | 1.68–5.59 | 1.13 | 0.40 | 0.728 | 0.57–2.26 |
| | 35–39 | 4.08 | 1.35 | <0.001 | 2.14–7.81 | 0.95 | 0.34 | 0.895 | 0.48–1.92 |
| | 40–44 | 3.97 | 1.52 | <0.001 | 1.88–8.41 | 1.02 | 0.38 | 0.949 | 0.50–2.11 |
| | 45–49 | 4.89 | 2.86 | 0.007 | 1.55–15.41 | 1.03 | 0.50 | 0.951 | 0.40–2.66 |

Others[a]: Widowed/divorced/separated, ANC: Antenatal Care, AOR: Adjusted odds ratio, SE: standard error

and the factors that have contributed to the skilled delivery during the last nine years in KA-HDSS sites. Results showed that the skilled delivery rate in 2017 was 96% (95% CI: 94.85%-97.05%), which was higher than the studies done in Ethiopia [14–16]. This might be due to the fact that in the current study area many interventions were implemented at different times, which could have increased access to a health facility and community awareness. This study revealed that, the trend of skilled delivery was significantly increased over time. The rate of skilled deliveries among reproductive age women was increased by 83% from 2009 to 2017. This might be due to the improved health service promotion and health service delivery. In addition, the strong referral linkage of pregnant women from community to health facilities could also increase the rate of skilled delivery. In this study, 35.89% mothers delivered at home. Of these, 97.87% were assisted by unskilled birth attendants. These women might face potential complications such as bleeding, retained placenta, ruptured uterus and infection which could lead to death. In the present study, in 2017 about 4% of the women delivered at home, which was lower than the study conducted in Gurage zone, Ethiopia [23]. The possible

reason could be in the current study area many interventions were implemented that could have increased access to health facility and community awareness on the benefits of healthcare services.

In the present study, the variables residence, marital status, educational status, occupation, and use of ANC service, were the determinants of the skilled delivery during the period of high skilled delivery rate (2014–2017). Primary education and secondary education and above were 1.63 and 2.1 times more likely to have skilled delivery service respectively as compared to those with no formal education. This finding was similar with the studies conducted in Ethiopia [20–22] where those who attended primary and secondary and above were more likely to utilize skilled delivery compared to those without formal education. This can be justified as education matters in knowledge acquisition and making a decision to utilize services. Women residing in urban areas increased the skilled delivery rate by 28 (AOR = 27.58; 95% CI: 3.86–196.97 as compared to rural residents. This is consistent with studies done in south and south west Ethiopia [21, 23]. This may be due to the fact that women residing in urban areas have more access to health information, access to nearby service and have more alternatives to health services compared to rural areas. Women with four or more ANC visits were 3.2 times more likely to have skilled delivery service than those having no ANC visits (AOR: 3.16; 95% CI: 2.33–4.28). These findings were similar to the previous studies conducted Ethiopia and Bangladesh [21, 23, 24]. This may be due to the educational packages given to ANC attendees that helped them to attend skilled delivery and postnatal care.

## Conclusions

The findings of this study showed the skilled delivery rate for the period of 2014–2017, was high. The trend of skilled delivery over the study period (2009–2017) showed a significant increase. The socio-demographic variables and use of ANC services were found to be statistically significant contributors to skilled delivery. Therefore, we recommend a balanced health information and access to health care that could address the huge discrepancy in skilled delivery.

## Acknowledgments

We would like to thank KA-HDSS office that provides permission with data access needed to conduct this research.

## Author Contributions

**Conceptualization:** Haftom Temesgen Abebe, Mache Tsadik Adhana, Mengistu Welday Gebremichael, Kebede Embaye Gezae, Assefa Ayalew Gebreslassie.

**Data curation:** Haftom Temesgen Abebe, Mache Tsadik Adhana, Mengistu Welday Gebremichael.

**Formal analysis:** Haftom Temesgen Abebe, Kebede Embaye Gezae, Assefa Ayalew Gebreslassie.

**Methodology:** Haftom Temesgen Abebe.

**Validation:** Haftom Temesgen Abebe, Mache Tsadik Adhana, Mengistu Welday Gebremichael, Kebede Embaye Gezae, Assefa Ayalew Gebreslassie.

**Writing – original draft:** Haftom Temesgen Abebe.

**Writing – review & editing:** Haftom Temesgen Abebe, Mache Tsadik Adhana, Mengistu Welday Gebremichael.

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
