## [Decision Letter · Decision Letter 0]

16 Apr 2021

PONE-D-21-07404

Magnitude, trends and determinants of skilled delivery from Kilite-Awlaelo Health Demographic Surveillance System, Northern Ethiopia, 2009- 2017

PLOS ONE

Dear Dr. Abebe,

Thank you for submitting your manuscript to PLOS ONE. After careful consideration, we feel that it has merit but does not fully meet PLOS ONE’s publication criteria as it currently stands. Therefore, we invite you to submit a revised version of the manuscript that addresses the points raised during the review process.

An expert in the field handled your manuscript, and we are appreciative for their time and thorough review. Although interest was found in your study, several major concerns arose that require your attention. 

We look forward to receiving your revised manuscript.

Kind regards,

Frank T. Spradley

Academic Editor

PLOS ONE

4. Please amend the manuscript submission data (via Edit Submission) to include author Haftom Temesgen.

5. Please amend your authorship list in your manuscript file to include author Haftom Abebe.

6. Please amend either the abstract on the online submission form (via Edit Submission) or the abstract in the manuscript so that they are identical.

7. Your ethics statement should only appear in the Methods section of your manuscript. If your ethics statement is written in any section besides the Methods, please delete it from any other section.

Reviewers' comments:

Reviewer's Responses to Questions

**Comments to the Author**

1. Is the manuscript technically sound, and do the data support the conclusions?

Reviewer #1: No

2. Has the statistical analysis been performed appropriately and rigorously? 

Reviewer #1: I Don't Know

3. Have the authors made all data underlying the findings in their manuscript fully available?

Reviewer #1: Yes

4. Is the manuscript presented in an intelligible fashion and written in standard English?

Reviewer #1: No

5. Review Comments to the Author

Reviewer #1: Access to care, and quality of the care provided at the facilities and the attending health care providers, are important factors determining pregnancy outcomes. Hence, this is an important issue to research and provide insight into how to improve the situation. The analysis and paper in its current form, however, do not add to the existing knowledge about factors that are associated with poor access to care and deliveries at health care facilities. Rather than trends the data could be analysed for various periods, say 2016, 2012, 2009 to identify the factors that were associated when the skilled delivery rates were low (in 2009) and factors that area associated when the skilled delivery rate has risen to above 90%. That might help identify strategies relevant for promoting skilled attendance at delivery for women who in 2020 still find it difficult to access health care facilities for delivery or decide to deliver at home for various socioeconomic, health system, geographic or cultural factors.

Abstract: The statements that “overall skilled delivery rate is 63.2%’ requires description/further explanation. Does it mean that the author added all the deliveries from 2009 to 2017 and 63.2% is the average across the years? If this is how the overall rate is calculated then it is unhelpful as what matters is the current rate of 96% and its comparison to the low rates a decade ago. An overall/average rate for all the deliveries across the years does not provide useful information.

Line 58-59: It is mentioned that the fundamental approach to improve maternal health is increasing skilled delivery. Access to skilled attendants at delivery is essential. However, it is important to consider the quality of care considering the recent research that points to poor outcomes despite high access. The data in this paper as well as a number of studies from around various developing countries inform that a majority of women is now delivered by the skilled attendants, and that a major contributory factor toward high maternal mortality is the poor quality of maternal health care. The quality of care is an increasingly fundamental concern and access to care and deliveries by skilled attendants need to be discussed within that context

Lines 67-69: Need to inform about the current MMR. At present 2011 MMR is presented. Since than access to skilled attendance has increased significantly. Hence, MMR o 2016 or 2017 is needed to be presented to contrast the 2009 and 2017 situation in terms of both access to skilled attendants at delivery and maternal mortality in Ethiopia.

Line 74-75: Not clear what is meant by “….respectively so the average skilled delivery rate of Ethiopia become 28% even if the health sector transformation plan of the country was set to be 90%”.

Line 92-101: Method that KA HDSS used needs to be presented in more detail, in order to better assess the methodology. Not clear what is meant by “retrospective open cohort study”.

METHOD: It seems over the years KA HDSS sample size increased from 14,455 households to 21,688 households. It is not described if 21,688 households included the same households or if every year the survey was conducted amongst a different set of households. If the survey is conducted amongst the same households there is need to discuss the potential impact of that research on health services utilisation by those households; respondents’ reflecting on health services access related questions in the survey might prompt for and improve healthcare seeking. Hence, there is a possibility that the respondents who take part in such repeated surveys act differently compared to the population at large. It is important to describe the methods of KA HDSS in more detail.

Lines 112-113: Some of the ‘births at home’ were conducted by skilled health care providers. If those deliveries are to be excluded then the study focus need to be ‘healthcare facility-based’ versus deliveries at homes.

Lines 148-149: Lack clarity, not clear what is meant by “…. the number of deliveries were 3,842(52.89%) observations had delivered one 148 times,2320(31.94%) two times, 966 (13.30%) three times,130(1.79%) four times and 5(0.07%) 49 five times”.

Lines 159-160: Lack clarity: “Based on their ANC visits, 8,515 of 11,925 (71.40%) of women had ANC visits at least one with a median time to visit was 2.5 times”.

Table 1/ 2: If data about ‘gravida’ is available it should be included and analysed.

Lines 177-178: Need to describe what is an ‘health extension worker’ (do they qualify as skilled attendants”).

Line 189-170: Need to explain what is meant by “overall skilled delivery rate”. The Figure 1 appears to be based on adding ALL deliveries across 9 years. Such averages are not helpful in defining the situation or trend. What matters is that in 2017 96% women delivered at the facilities.

Lines 194-195: Need clarity. “The trend of ANC attendance in the study period (2009-2017) showed a significant change, increased from 48.44% (95% CI: 42.72%-54.21%) in 2009 to 94.7% (95%CI: 93.31-95.83%) in 2017. One ANC or Four ANC?

Lines 193-207: ANC, HIV, malaria bed nets use are not the focus of this study. There are considered as independent variable for this study which attempts identifying the factors that are associated with accessing skilled birth attendants at health facilities. For this paper it is sufficient to inform if these were or were not associated with having delivery by a skilled attendants. Information on these variables per se is not relevant. For the same reasons, figures 3, 4, 5 are not needed.

DISCUSSION needs to be strengthened. At present many of the results are repeated in the discussion, and the discussion does not discuss in depth to add to the existing knowledge about factors that determine access to skilled attendants at delivery/births at health facilities.

If the data is reanalysed in light of the suggestion above, the discussion would need to be aligned with the revised analysis and results.

There are significant difference noted between access to skilled attendants/ delivery at facilities between urban and rural areas. This finding needs to be discussed within the context of urban and rural MMR in Ethiopia with reflections whether better access in urban areas translates into better pregnancy outcomes and lower maternal mortality ration.

REFERENCES needs to be correctly formatted in line with the Journal requirement.

ENGLISH LANGUAGE editing is required. Some examples of statements that lack clarity include:

Line 50: ‘The skilled attendant is an accredited health professional of midwives, doctors, and nurses with midwifery and life-saving skills’. What is meant by ‘an accredited health professional of midwives……’?.

line 219-220: “Like wisely, single women who gave their recent birth from 2009 –2017 in the study setting were 2.13 (AOR: 2.13; 95% CI: 1.71 –2.65) times more likely to have skilled delivery

Line 255: “….higher than the studies done in Ethiopia [16-19].This might be related to the fact that in the current study area many interventional have been implemented…”

Line 276: “….. that women residing in urban areas are more accessible to health information, access…”

6. PLOS authors have the option to publish the peer review history of their article (what does this mean?). If published, this will include your full peer review and any attached files.

Reviewer #1: No

---

## [Author Response · Author response to Decision Letter 0]

28 May 2021

Manuscript ID: PONE-D-21-07404. 

Magnitude, trends and determinants of skilled delivery from Kilite-Awlaelo Health Demographic Surveillance System, Northern Ethiopia, 2009- 2017

Authors: Abebe et al.

Dear Professor Frank T. Spradley,

First of all we would like to thank you very much for the chance you gave us to revise the manuscript and we really appreciate the comments and suggestions. 

Based on the instructions provided in the journal's website and your email on April 16, 2021, we have revised the manuscript along the line of all comments made by the academic editor and reviewer. 

Appended to this letter is our point-by-point response to the comments raised by the academic editor and reviewer. Essentially, we agreed with almost all the comments, and we would like to express our sincere thanks to the referee for identifying areas of our manuscript that needed modifications or corrections. We would also like to thank you for allowing us to re-submit a revised version of the manuscript.

We hope that the revised manuscript is acceptable for publication in PLOS ONE.

Sincerely Yours,

Haftom Temesgen Abebe 

Responses to academic editor comments

 Response: We have revised the manuscript according the PLOS ONE’s style. (see the revised manuscript)

Response: Thank you for the suggestions. Our paper is edited by Dr Carmer C. Robles (PhD), email: chenyta08@yahoo.com

Response: The data has potentially identifying information including name and households number. The data can be obtained from the institutional office Kilite-Awlaelo Health Demographic Surveillance System (KA-HDSS), College of Heath Science, Mekelle University, Email: ka.hdss.2011@gmail.com; Tel: +251914743841

4. Please amend the manuscript submission data (via Edit Submission) to include author Haftom Temesgen.

Response: Done, we have now corrected this.

5. Please amend your authorship list in your manuscript file to include author Haftom Abebe. 

Response: Done.

6. Please amend either the abstract on the online submission form (via Edit Submission) or the abstract in the manuscript so that they are identical.

 Response: Done.

7. Your ethics statement should only appear in the Methods section of your manuscript. If your ethics statement is written in any section besides the Methods, please delete it from any other section.

Response: Done. Please see the revised paper.

Responses to reviewers’ comments 

First of all we would like to thank you very much for the comments.

Reviewer1:

Access to care, and quality of the care provided at the facilities and the attending health care providers, are important factors determining pregnancy outcomes. Hence, this is an important issue to research and provide insight into how to improve the situation. The analysis and paper in its current form, however, do not add to the existing knowledge about factors that are associated with poor access to care and deliveries at health care facilities. Rather than trends the data could be analysed for various periods, say 2016, 2012, 2009 to identify the factors that were associated when the skilled delivery rates were low (in 2009) and factors that area associated when the skilled delivery rate has risen to above 90%. That might help identify strategies relevant for promoting skilled attendance at delivery for women who in 2020 still find it difficult to access health care facilities for delivery or decide to deliver at home for various socioeconomic, health system, geographic or cultural factors.

Response: Thank you very much for the suggestion. We have now identify the factors associated with the skilled delivery for the period of low skilled delivery rates (2009-2011) and factors that were associated with the outcome for the period of high skilled delivery rate(2014-2017) (Table 3). 

Note that because of zero cells for most of the variables we could not analysed separately for various periods (2009, 2017), instead we analysed for 2009-2011 when the skilled delivery rate was low and for the period 2014-2017 when the skilled delivery rates was above 82%. Please see the revised manuscript Table 3. 

Abstract: The statements that “overall skilled delivery rate is 63.2%’ requires description/further explanation. Does it mean that the author added all the deliveries from 2009 to 2017 and 63.2% is the average across the years? If this is how the overall rate is calculated then it is unhelpful as what matters is the current rate of 96% and its comparison to the low rates a decade ago. An overall/average rate for all the deliveries across the years does not provide useful information.

Response: Yes, the overall skilled delivery rate is the average across the years. We have now removed this. Please see the revised paper

Line 58-59: It is mentioned that the fundamental approach to improve maternal health is increasing skilled delivery. Access to skilled attendants at delivery is essential. However, it is important to consider the quality of care considering the recent research that points to poor outcomes despite high access. The data in this paper as well as a number of studies from around various developing countries inform that a majority of women is now delivered by the skilled attendants, and that a major contributory factor toward high maternal mortality is the poor quality of maternal health care. The quality of care is an increasingly fundamental concern and access to care and deliveries by skilled attendants need to be discussed within that context

Response: Thank you for the comments. We have now included. Please see the revised. 

Lines 67-69: Need to inform about the current MMR. At present 2011 MMR is presented. Since than access to skilled attendance has increased significantly. Hence, MMR o 2016 or 2017 is needed to be presented to contrast the 2009 and 2017 situation in terms of both access to skilled attendants at delivery and maternal mortality in Ethiopia.

Response: Thank you for the Comments. We have now included MMR for 2016. See the revised paper.

Line 74-75: Not clear what is meant by “….respectively so the average skilled delivery rate of Ethiopia become 28% even if the health sector transformation plan of the country was set to be 90%”.

Response: We agree and we have now revised this statement. Please see the revised manuscript.

Line 92-101: Method that KA HDSS used needs to be presented in more detail, in order to better assess the methodology. Not clear what is meant by “retrospective open cohort study”.

Response: Thank you again for the comments. We have now described this in detail. Please see the revised. 

METHOD: It seems over the years KA HDSS sample size increased from 14,455 households to 21,688 households. It is not described if 21,688 households included the same households or if every year the survey was conducted amongst a different set of households. If the survey is conducted amongst the same households there is need to discuss the potential impact of that research on health services utilisation by those households; respondents’ reflecting on health services access related questions in the survey might prompt for and improve healthcare seeking. Hence, there is a possibility that the respondents who take part in such repeated surveys act differently compared to the population at large. It is important to describe the methods of KA HDSS in more detail.

Response: The 21,688 households include existing households (14, 455 HHs) and newly households that has been included later in 2016. We have now described this in more detail. Please see the revised. 

Lines 112-113: Some of the ‘births at home’ were conducted by skilled health care providers. If those deliveries are to be excluded then the study focus need to be ‘healthcare facility-based’ versus deliveries at homes.

Response: Thank you for the comments. Indeed some of the births at home were delivered by skilled birth attendants. We have now corrected this, i.e., the outcome was dichotomizes as one if a women gave birth by skilled birth attendants and 0 otherwise. Accordingly we have reanalysed and modified our results (Table 1, Table 2 and Figure 3). Please see the revised paper.

Lines 148-149: Lack clarity, not clear what is meant by “…. the number of deliveries were 3,842(52.89%) observations had delivered one 148 times,2320(31.94%) two times, 966 (13.30%) three times,130(1.79%) four times and 5(0.07%) five times”.

Response: Comments accepted. We have now corrected. Please see the revised manuscript. 

Lines 159-160: Lack clarity: “Based on their ANC visits, 8,515 of 11,925 (71.40%) of women had ANC visits at least one with a median time to visit was 2.5 times”.

Response: We have now revised and corrected the statement. 

Table 1/ 2: If data about ‘gravida’ is available it should be included and analysed.

Response: Thank you for the suggestion. We have now included this in Table 2. Please see in the revised paper.

Lines 177-178: Need to describe what is an ‘health extension worker’ (do they qualify as skilled attendants”).

Response: No they don’t qualify as skilled birth attendants. As defined by WHO skilled birth attendant is a health professional such as midwife, doctor or nurses who has been educated and trained to proficiency in the skills needed to manage women during normal (uncomplicated) childbirth and the immediate postnatal period as well as in the identification. 

We have now described this. Please see the revised paper.

Line 189-170: Need to explain what is meant by “overall skilled delivery rate”. The Figure 1 appears to be based on adding ALL deliveries across 9 years. Such averages are not helpful in defining the situation or trend. What matters is that in 2017 96% women delivered at the facilities.

Response: Yes, the overall skilled delivery rate is the average across the years. We agree and we have now removed this. Please the revised paper

Lines 194-195: Need clarity. “The trend of ANC attendance in the study period (2009-2017) showed a significant change, increased from 48.44% (95% CI: 42.72%-54.21%) in 2009 to 94.7% (95%CI: 93.31-95.83%) in 2017. One ANC or Four ANC?

Response: The comment is accepted. This trend is for ANC attendance at least once. We have described this. Please see the revised paper.

Lines 193-207: ANC, HIV, malaria bed nets use are not the focus of this study. There are considered as independent variable for this study which attempts identifying the factors that are associated with accessing skilled birth attendants at health facilities. For this paper it is sufficient to inform if these were or were not associated with having delivery by a skilled attendants. Information on these variables per se is not relevant. For the same reasons, figures 3, 4, 5 are not needed.

Response: We agree and we have now dropped the figures (4 and 5). Please see the revised manuscript.

DISCUSSION needs to be strengthened. At present many of the results are repeated in the discussion, and the discussion does not discuss in depth to add to the existing knowledge about factors that determine access to skilled attendants at delivery/births at health facilities.

Response: We have now revised the discussion. Please see the revised paper. 

If the data is reanalysed in light of the suggestion above, the discussion would need to be aligned with the revised analysis and results.

Response: We have revised the discussion based on the suggestions. Please see the revised paper. 

There are significant difference noted between access to skilled attendants/ delivery at facilities between urban and rural areas. This finding needs to be discussed within the context of urban and rural MMR in Ethiopia with reflections whether better access in urban areas translates into better pregnancy outcomes and lower maternal mortality ration.

Response: Comment accepted. We have now done this. Please see the revised manuscript.

REFERENCES needs to be correctly formatted in line with the Journal requirement.

Response: We have the corrected the reference formatted in line with the Journal requirement. 

ENGLISH LANGUAGE editing is required. Some examples of statements that lack clarity include:

Line 50: ‘The skilled attendant is an accredited health professional of midwives, doctors, and nurses with midwifery and life-saving skills’. What is meant by ‘an accredited health professional of midwives……’?.

line 219-220: “Like wisely, single women who gave their recent birth from 2009 –2017 in the study setting were 2.13 (AOR: 2.13; 95% CI: 1.71 –2.65) times more likely to have skilled delivery

Line 255: “….higher than the studies done in Ethiopia [16-19].This might be related to the fact that in the current study area many interventional have been implemented…”

Line 276: “….. that women residing in urban areas are more accessible to health information, access…”

Response: Thank you again. We have now edited the English language by colleague Dr Carmer C. Robles (PhD), email: chenyta08@yahoo.com. Please see the revised paper.

---

## [Editor Report · Decision Letter 1]

21 Jun 2021

Magnitude, trends and determinants of skilled delivery from Kilite-Awlaelo Health Demographic Surveillance System, Northern Ethiopia, 2009- 2017

PONE-D-21-07404R1

Dear Dr. Abebe,

We’re pleased to inform you that your manuscript has been judged scientifically suitable for publication and will be formally accepted for publication once it meets all outstanding technical requirements.

Kind regards,

Frank T. Spradley

Academic Editor

PLOS ONE

---

## [Editor Report · Acceptance letter]

25 Jun 2021

PONE-D-21-07404R1 

Magnitude, trends and determinants of skilled delivery from Kilite-Awlaelo Health Demographic Surveillance System, Northern Ethiopia, 2009- 2017 

Dear Dr. Abebe:

I'm pleased to inform you that your manuscript has been deemed suitable for publication in PLOS ONE. Congratulations! Your manuscript is now with our production department. 

Kind regards, 

on behalf of

Dr. Frank T. Spradley 

Academic Editor

PLOS ONE